# Comparative Measurement of Horizontal Penetrometry with a Focus on the Degree of Soil Compaction in Real Work Conditions

**Marek Mojžiš, Ján Kosiba and Ján Jobbágy ***

Faculty of Engineering, Institute of Agricultural Engineering, Transport and Bioenergetics, Slovak University of Agriculture in Nitra, Tr. A. Hlinku 2, 949 76 Nitra, Slovakia; xmojzism@uniag.sk (M.M.); jan.kosiba@uniag.sk (J.K.)
\* Correspondence: jan.jobbagy@uniag.sk

**Abstract:** Potential soil production is closely related to the physical and mechanical properties. The aim of this paper was to evaluate the effect of different levels of soil compaction created by tractor chassis. The total area of the experimental plot was 13.22 ha. Up until 2019, a conventional tillage system had been used. The measurements were carried out with an innovative measuring device that allows for the continuous measurement of the horizontal penetrometry for comparative measurements while driving, which was designed at the Slovak University of Agriculture in Nitra. The measuring device measured the soil resistance in the tire track (On-track) and out of track (Off-track) as well as in three (50 s) sequences within one tractor pass. Three lines were chosen, where in each a pair of combinations was made. The results were subjected, in addition to graphical evaluation, to single factor ANOVA analysis. When comparing individual passes (PH1 to PH6), the statistical analysis showed that the results of the horizontal resistance measurements proved to be statistically significant ($p < 0.05$) with respect to the weight, number of passes, and tire underinflation. The highest compaction was caused by the first pass, while the higher weight of the tractor during the next pass had a smaller effect. Underinflating the tires ensured a reduction in compaction. Reducing the tractor tire pressure to 0.15 MPa resulted in a reduction in soil compaction of up to 16%.

**Keywords:** horizontal penetrometric resistance; soil; compaction; agriculture





## 1. Introduction

The soil is a place for growing crops, and fruits are an essential source of nutrition for humanity. With the development of automation, modern technology has taken the direction of increased work efficiency, a fact that is also linked to increasing weight. In many developing countries, the phenomenon of increased soil compaction is consequently noticeable for this reason.

More than 30 years ago, a global semi-quantitative assessment of soil degradation reported that approximately 33 Mha in Europe was affected by soil compaction [1], equivalent to about one third of all arable land in Europe. Additionally, the weight of agricultural machinery is steadily increasing, for example, wheel loads on combine harvesters increased by approximately 65% between 1989 and 2009 [2]. As a consequence, the mechanical stresses exerted by today's machinery can create the disproportionate compaction of arable soils [2–4].

Soil compaction has become one of the most important issues in modern agriculture. With increasing demands on the efficiency of work and the performance of the equipment used, the weight of the equipment is increasing. This, combined with incorrect use and the inappropriate organization of the movement of the machine through the fields, often results in extreme soil compaction. Excessive soil compaction results in reduced yields, which means that the compaction has to be eliminated, usually by highly energy-intensive work operations. Soil compaction is also considered to be one of the causes of yield stagnation observed in many European countries [5].

Soil strength can be defined as the resistance of the soil that must be overcome when the soil structure is changed to achieve deformation. In soil processing operations, high soil strength can be both an advantage and a disadvantage, depending on the circumstances. It is an advantage when it allows movement through the soil as its bearing capacity is increased, but it is also a disadvantage in soil working as it increases resistance and makes it more difficult to achieve a suitable soil structure when working as well as impairing seed germination and root growth [6].

Nowadays, basically all technological operations of field crop production are provided by agricultural machinery. This brings benefits such as a higher efficiency or lower production cost per hectare of land processed. However, one of the main side effects is the soil compaction caused by the passage of agricultural machinery. This is a very important factor, which, according to Khaledian et al., (2017), results in an accelerated loss of carbon and nitrogen from the cultivated soil and thus contributes significantly to the degradation of an already stressed environment [7].

Compacted soil requires more power and energy to work the soil. It causes a tenfold to sixteenfold increase in the energy required at low speeds and a fourfold to eightfold increase at high speeds. The pull from the narrow chisel points has increased from 31 kg in uncompacted soil to 159 kg in compacted soil. Heavy axle loads will increase the depth of compaction in the soil profile. As loads increase above 10 tons per axle, the potential to compact the soil down to below the cultivation layers also increases. Large 500 horsepower tractors, combines, tankers, and grain transloaders can weigh from 18 to 40 tons per axle and can create a compaction of up to 60 to 90 cm, regardless of whether they are equipped with tracked chassis or tires. If the axle loads do not exceed 10 tons, the compaction is generally localized in the topsoil to a depth of 15 to 20 cm [8].

It is clear from both practical and scientific experience and knowledge that soil processing must be carried out at a suitable soil moisture range, which will also ensure the negative impact of the input of heavy machinery, namely soil compaction. In real-time, however, the economic consequences of delays in sowing, harvesting, or other operations are important.

Options for minimizing soil compaction are based on basic knowledge and it is not advisable to enter the field unless the moisture conditions are suitable. In addition, it is advisable in practical conditions to reduce the number of passes across the field, reduce the weight of machinery and mechanization equipment, choose the right chassis, possibly the right tire inflation, and last but not least, to change the organization of the movement of machinery across the fields, namely CTF [9].

Based on the above information and knowledge, we decided to investigate the soil compaction monitored under realistic conditions, simulating several different input passes. Therefore, the following hypotheses can be stated:

➢ The highest compaction is achieved in the first pass;
➢ Each additional pass reduces compaction;
➢ Tire pressure affects the soil compaction.

## 2. Materials and Methods

### 2.1. The Characteristics of the Experimental Plot and the Farm

The experimental plot is located in the cadastre of the village Hronské Kosihy (Figure 1). It has been managed by Monika Mojžišová SHR for more than 23 years. The total area of the experimental plot is 13.22 ha. Up until 2019, the conventional tillage system with medium-depth ploughing every other year was used. According to the soil maps of the classification of bonitated soil-ecological units (BPEJ, Slovak national standard for identification of soil production potential), the land is predominantly composed of two soil types: (1) clay-loam soil type, typical medium heavy 0006002, and (2) black soil type 0019002. More detailed information about the farm and the farming techniques used has been presented in another paper [10].

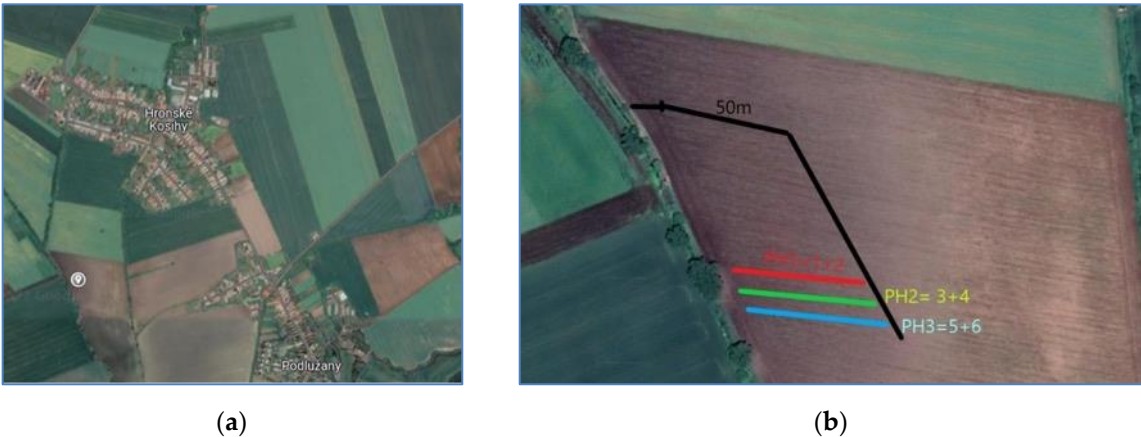

**Figure 1.** Experimental plot and area of interest: (**a**) focused plot; (**b**) location of measurement.

In the year when all field measurements were taken, winter wheat was grown. The experimental area is used for a larger study of the effects of farm equipment passes on the soil and was targeted with a Leica GPS. The organization of the movement of the machinery was pre-arranged and planned. On this part of the experimental plot, deeper processing of the stubble was carried out in order to create a homogeneously processed topsoil for the following measurements and detect differences in soil compaction (depth of 10 to 15 cm, the overlap of the individual working passes was more than 100%).

## 2.2. Penetrometric Soil Resistivity—Horizontal Direction

The equipment used for this measurement was specially designed at the Technical Faculty of the Slovak University of Agriculture in Nitra (Figure 2). It allows for the measuring and recording of a large amount of data continuously while moving along the measured plot. The device is compatible with most of the commonly used three-point tractor hitches and is equipped with a pair of knives, where it is possible to change both the setting of the measuring knives and their mutual distance [11]. The accuracy of the sensor EMS 150 is 0.5, etc.; the failure is ±5 N. A more detailed description of the measuring device is also given in that paper [11].

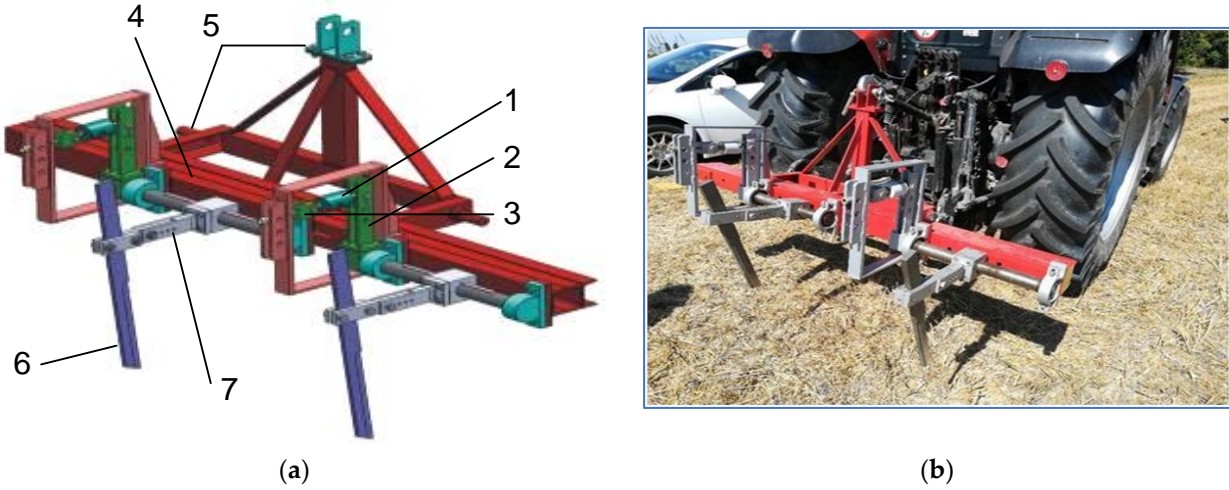

**Figure 2.** Measuring instrument. (**a**) Measuring instrument with two-element comparison method 1—power sensor EMS 150, 2—arm, 3—attachment to the power sensor, 4—frame, 5—attachment of the three-point hitch, 6—blade, 7—blade clutch. (**b**) Measurement of the horizontal soil resistivity in comparative measurements.

This makes it possible to perform comparative measurements, the results of which take into account the difference in the measured data on and off the track, which were formed by the tractor tire. This figure expresses the increment of force due to overrunning. Such an expression is only possible in the case of a comparative measurement where the second knife senses the force outside the track in the immediate vicinity in uncompacted soil.

Experimental measurements using a device for detecting soil resistance in the horizontal direction (the force required to cut through the soil layer with a knife sunk to a set depth) were carried out according to the following principles:

➢ Measurements were carried out in purposefully modeled machine tracks:

 ✓ Direct measurements (measurement with one knife in a purposefully tamped footprint)—six different soil compaction models + control;
 ✓ Measurements in extreme conditions.

➢ The experiments were carried out in real production conditions on a selected plot with selected machines and machine sets.

➢ Compaction modeling was carried out by driving tractors and implemented over the soil:

 ✓ In real conditions during cereal harvesting with predetermined movements of machines and machine implements;
 ✓ For each type of experimental run, at least 3 repetitions were performed;
 ✓ The time of each measurement was 50 s (corresponding to a distance of approximately 28 m);
 ✓ During the experiment, a steady speed of 2 $\text{kmh}^{-1}$ was maintained;
 ✓ Write frequency of 0.02 s, (approx. 2500 data per trip);
 ✓ Measurements started in the direction parallel to the track lines at a distance of 30 m from the ditch (avoiding the influence of the measurement results, for example, by incorrect turning of the machinery during crop treatment, etc.); the same preset positions of the three-point linkage of the tractor used as a carrier for the horizontal penetrometer were always used.

➢ Evaluation of the measured data: Selection of the exact section from each measurement.

The results of the soil resistance measurements after several tractor passes were recorded by HYDAC 2020 and processed using a PC.

Measurements were made in three sections (PH1, PH2, and PH3; Figure 1) with two measurements in each section. The first measurement (Table 1) was carried out in the first line with a Zetor Forterra 135 tractor (ZETOR TRACTORS a. s., Brno, Czech Republic) in combination with the measuring set up without prior compaction (first pass and measurement at that moment). The second measurement (second pass) was carried out in the same track after the first pass of the Zetor Forterra 135 tractor with the measuring device (i.e., the second pass of the Zetor Forterra 135 tractor with the measuring device). In the second line, one pass with a Valtra T202 tractor (VALTRA Inc., Suolahti, Finland) in combination with weights and carried implements (discs) was applied before the first horizontal penetrometry measurement (PH2.3). The tire pressure was set to the prescribed value (0.19 MPa). The fourth measurement was carried out again in the second line (PH2.4) with the Valtra T202 tractor passing through the line twice and the Forterra 135 tractor once, in combination with the above-mentioned tools or equipment. The fifth measurement (PH3.5) was carried out in a similar way to the third, but with the difference that we focused on tire underinflation (0.15 MPa). The last measurement was similar to the fourth, also with the same tire underinflation. Thus, the aim was also to investigate the effect of changing the tractor passes or the underinflation of the tires on the resulting soil compaction. The work activities are detailed in Table 2.

**Table 1.** Technical parameters of the machines in the measurements.

| Experimental Measurement | Marking | Soil Moisture Content, %-Weight | Applied Set with Total Weight m (kg), Number of Passes $p$ (1× or 2×) | Tire Inflation Pressure (×100 kPa) |
|---|---|---|---|---|
| 1 | PH1.1 | 19.19 | Zetor + measuring device (m = 6010), $p$ = 1× | 1.8 |
| 2 | PH1.2 | 19.19 | Zetor + measuring device (m = 6010), $p$ = 2× | 1.8 |
| 3 | PH2.3 | 17.23 | Valtra + carried tools (m = 10,535), $p$ = 1× | 1.9 |
| | | | Zetor + measuring device (m = 6010), $p$ = 1× | 1.8 |
| 4 | PH2.4 | 17.23 | Valtra + mounted attachment (m= 10,535), $p$ = 2× | 1.9 |
| | | | Zetor + measuring device (m = 6010), $p$ = 2× | 1.8 |
| 5 | PH3.5 | 15.61 | Valtra + mounted attachment (m = 10,535), $p$ = 1× | 1.5 |
| | | | Zetor + measuring device (m = 6010), $p$ = 1× | 1.8 |
| 6 | PH3.6 | 15.61 | Valtra + mounted attachment (m = 10,535), $p$ = 2× | 1.5 |
| | | | Zetor + measuring device (m = 6010), $p$ = 2× | 1.8 |

**Table 2.** Work activities and run breakdown with the resistance results.

| Marking | Number of Passes in the Track | Number of On-Track and Off-Track Measurements | VK On-Track | VK Off-Track | Ø R On-Track, N | Ø R Off-Track, N |
|---|---|---|---|---|---|---|
| PH1.1 | 1 times MS | 25,010 | 19.81 | 32.51 | 2712.67 | 2134.64 |
| PH1.2 | 2 times MS | 15,006 | 16.71 | 38.69 | 2980.32 | 1230.14 |
| PH2.3 | 1 times V + 1 times MS | 15,006 | 15.81 | 24.80 | 3119.23 | 2539.18 |
| PH2.4 | 2 times V + 2 times MS | 12,062 | 19.72 | 27.62 | 3180.87 | 1685.17 |
| PH3.5 | 1 times V + 1 times MS | 15,006 | 20.61 | 32.87 | 2778.87 | 1585.75 |
| PH3.6 | 2 times V + 2 times MS | 15,006 | 22.00 | 40.62 | 2746.84 | 1413.31 |

MS—measuring set, V—Valtra tractor, VK—coefficient of variation, R—penetrometric resistance.

## 3. Results

Within the research activities focused on soil compaction, we focused on the application of the measuring device designed at the SPU in Nitra. The device has already been applied at different input settings, and the results have been published in various journals (e.g., [11]). Prior to our measurements, we first implemented a stubble treatment where we uniformly tilled the topsoil at a prescribed depth. During the experimental measurements, a 20 cm deepening of a pair of knives was used, which was conducted under six different experimental variants (PH1.1 to PH3.6). The results of the resistances in each line are presented in Table 2.

Table 1 shows the applied experimental measurements of the horizontal penetrometry. For all measurements listed, these were performed by a pair of knives, and the simultaneous measured values from the two strain gauges were evaluated together. Potential hypotheses investigated were based on the variation in the number of passes and tire pressure and in their ultimate effect on soil compaction. The method of knife placement on the applied equipment was based on the equipment itself, and in our case, these were comparative measurements, with one knife placed centrally between the tractor wheels and the other

placed behind the tractor wheel (Figures 2 and 3). The recording of the measured data in terms of the methodology was evaluated separately for each experimental pass.

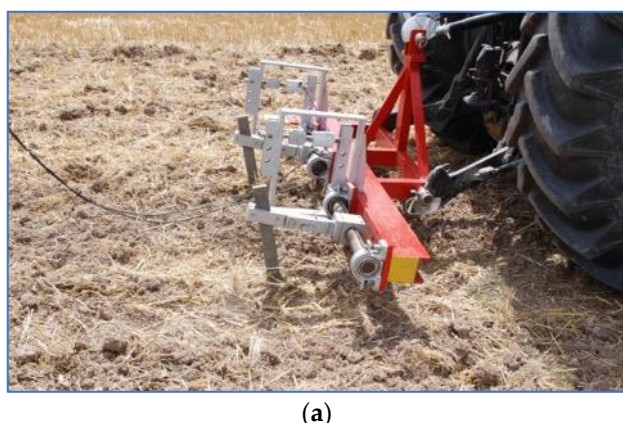

(**a**)

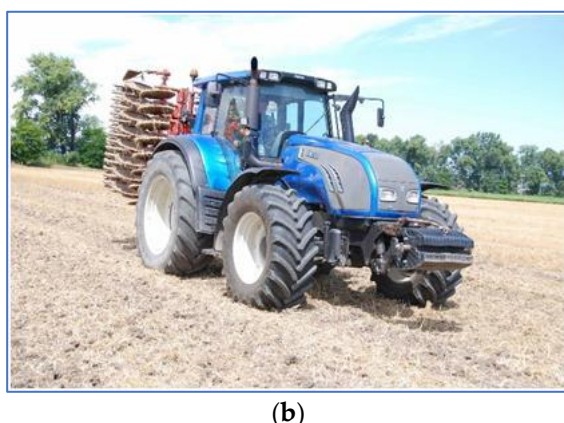

(**b**)

**Figure 3.** Field measurements. (**a**) Depth of the horizontal penetrometer blades (field measurement). (**b**) Valtra T202 tractor with Kverneland Qalidisc discs (Kverneland Group, Klepp, Norway).

**Measurement PH1.1.** In this practical measurement, we focused on the assessment of soil compaction in the area where the soil was compacted only by passing the tractor itself with the measuring equipment directly during the measurement. Therefore, the land was uniformly treated with a disc compactor with the same settings over the entire area of the experimental measurement. The average soil moisture content in this line was 19.19%. A graphical representation of the results for the PH1.1 variant of the horizontal penetrometry run is shown in Figure 4. In total, the measurement was carried out for 250 s, during which we were able to obtain up to 12,505 soil resistivity data records. In terms of the distribution of the measured penetrometric resistivity values, this was a normal distribution. The average value in the tractor wheel track reached a resistance of 2712.67 N, while outside the track it was 2134.64 N. However, the value of the coefficient of variation (CV) was significantly different, reaching 19.81% in the track and 32.51% outside the track. The soil moisture content averaged 19.19%. From the graphical progression, we can see that in some places during the measurements, we encountered an obstacle that induced extreme values (outside the footprint—value higher than 4.000 N). Within the investigated section, the values on-track ranged from 1154.9 N up to 4588.14 N, which represents an increase of up to 3.97 times. The standard deviation (SD) was 537.33 N on-track, and 693.99 N off-track. The off-track measurements represented a range of measured resistivity of 4131.11 N with a total overshoot of up to 10.47 times between the minimum and maximum values (4566.9 N). Statistical analysis across the entire reach demonstrated significance, which means that the on-track and off-track results were significantly different ($p < 0.05$, F = 5423.7), so we focused on a more detailed investigation of the studied section. In terms of measurements, we had the total time segment divided into five 50 s segments. The plot further shows that during the first 50 s, the differences in the on-track and off-track values were not significant (F = 1.25, $p = 0.26$; no statistically significant differences between the two groups of values, namely on-track and off-track, were demonstrated, analysis terminated). The mean value was 2593.38 N (SD = 303.65 N) on-track and 2607.71 N (SD = 561.95 N) off-track. In that section, the resistances in the trace and off-track alternated, that is, they were higher in the trace in some sections and off-track in others. This may have been due to soil variability but also to inappropriate organization of crossings during harvesting and other working operations in the area. In the past, during harvesting operations on this parcel, in order to be more efficient, the combine harvester's grain bin was emptied on the move into the haul trucks intended for road applications; this caused deep compaction that probably could not be completely eliminated. However, the evaluation of a further 50 s section showed statistically significant differences ($p = 0.54.10$–3; F = 11.98, F > Fcrit). The mean value in that case was 2561.38 N (SD = 513.86 N) inside the trace and

2508.91 N (SD = 557.51 N) off-track. Thus, for the third 50 s segment, the results also proved to be statistically significant ($p = 0$). The mean value of horizontal resistance was 3040 N (SD = 576.7) on-track and 1638.84 N (SD = 576.17 N) off track. The fourth trace at 50 s also showed significant differences (F = 1627.38 > Fcrit). The mean value of horizontal drag on-track was 2746.46 N (SD = 494.75 N), and off-track it was 2085.22 N (SD = 653.59 N). Within the section, the on-track and off-track values were close to each other, with the off-track resistance values exceeding the on-track values in places. However, the highest values were in the first 50 s section of the measurement that was closest to the edge of the plot. We know that this part of the plot has in the past been subjected to, for example, the turning of harvesting rigs during harvesting operations, so the compaction caused by this has clearly not been eliminated. For these reasons, the results of some parts may be illogical, but these are measurements under real production conditions and are variable, as can be seen in the following graph.

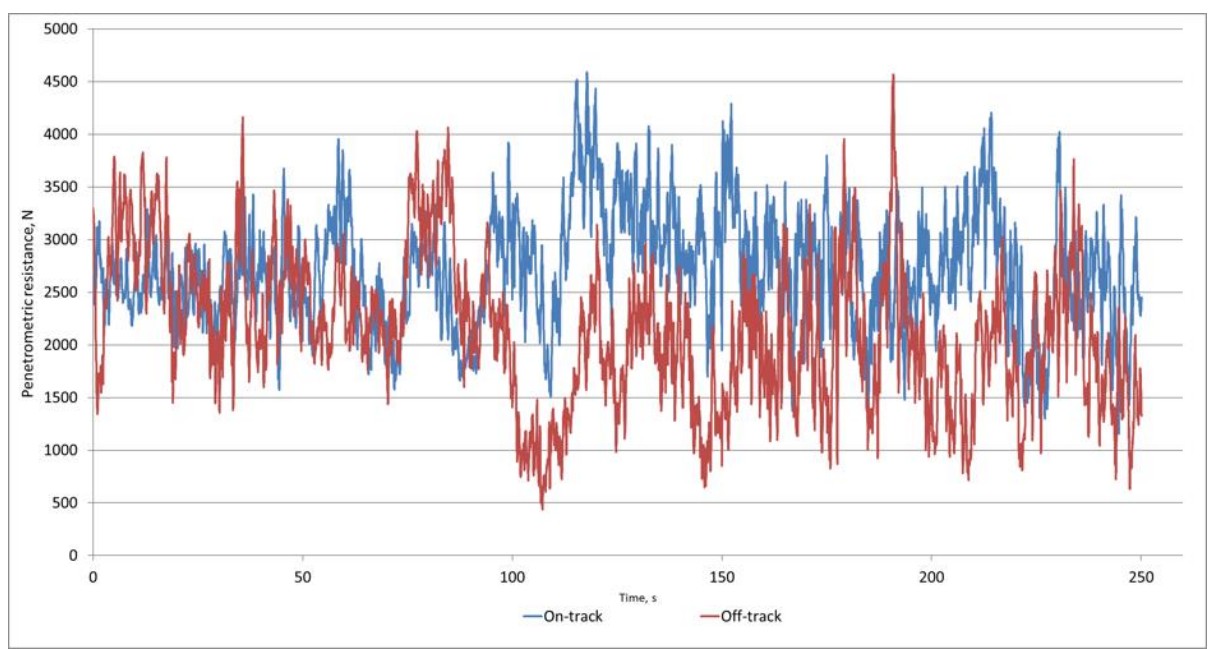

**Figure 4.** Comparative measurement, PH1.1, conditioned.

**Measurement PH1.2.** For the next measurement, we decided to repeat the measurements with two passes with a Zetor Forterra tractor in combination with the measuring equipment. A graphical representation of the results (variant PH1.2) of the horizontal penetrometry is shown in Figure 5. The average value in the tractor footprint reached a higher value (only by 267.65 N) than in the first measurement (2980.32 N, Table 2, SD = 498 N). However, the overall graphical progression shows that the level of dependence at the significance level (0.05), namely the influence of on- and off-track compression, was significantly higher ($p < 0.05$, F was higher compared to Fcrit up to 12,605.5 times). The mean value of horizontal resistance outside the tractor footprint was 1230.14 N (SD = 475.95 N, the value of VK was 38.69%). The values of the coefficients of variation were slightly different (on-track—16.71%, off-track—38.69%). However, the number of records was up to 1.67 lower in that experiment. Thus, it was demonstrated that the tractor passage (PH1.1) had an effect on soil compaction. The measurements outside the tractor tracks were interesting, where the mean value decreased up to 1.74 times in the second pass. This could also have been due to the first pass where we disturbed the soil with the measuring device outside the tractor wheel tracks. Statistical analysis showed that the horizontal resistance measurements and the comparison of two passes of the Zetor tractor with the measuring device with one pass were proven to be statistically significant ($p < 0.05$), although the value of the mean value was only 1.09 times higher.

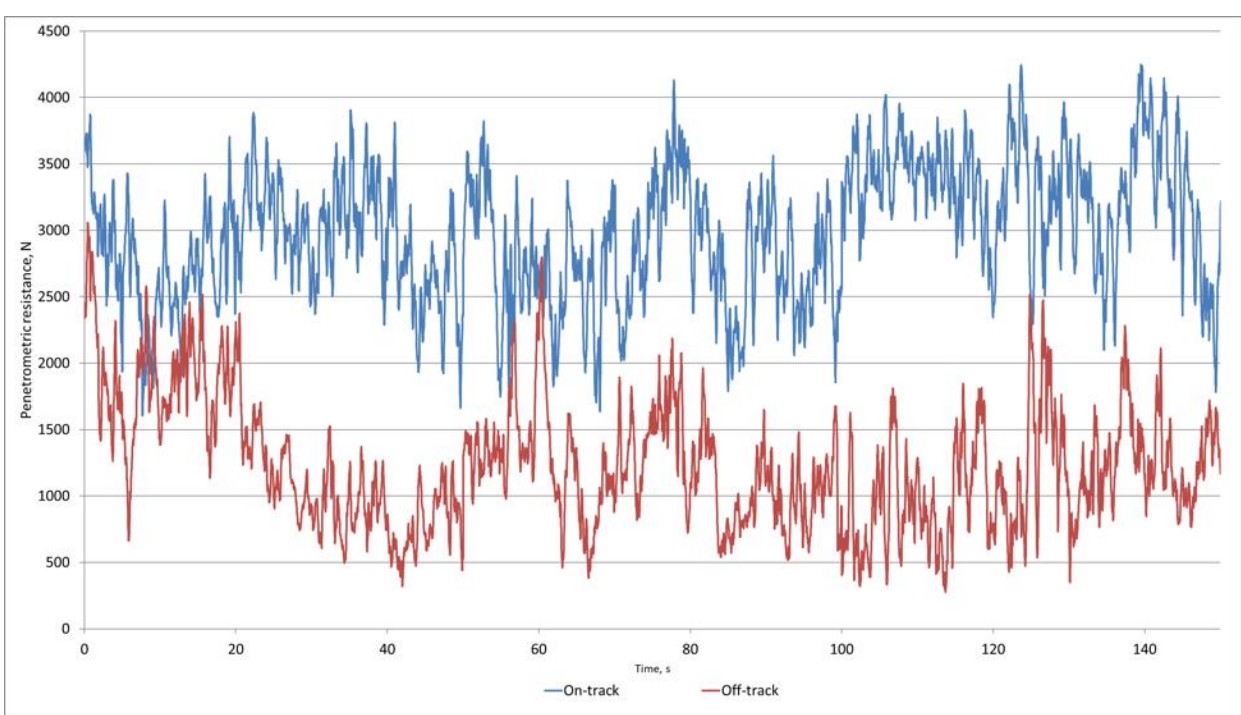

**Figure 5.** Comparative measurement, PH1.2, conditioned.

**Another pair of measurements (PH2.3 and PH2.4)** was performed under altered input conditions (Figure 1). The measurements were shifted by a few meters within the investigated terrain, and the average soil moisture content in this line was 17.23%. The difference from the first pair of measurements was that a single pass with a Valtra T202 tractor with Kverneland Qalidisc discs and a total weight of 10.535 t was applied on the given research route PH2.1 (the tractor was loaded with 1000 kg of weights in the front three-point hitch). The tire pressure of the said tractor was 0.19 MPa and it was fitted with Alliance agri-star365 tires with the size 540/65 R30 on the front axle and Michelin multibib 650/70 R42 tires on the rear axle. Subsequently, a Zetor Forterra tractor (total weight 6.14 t) was used for the crossing in combination with the horizontal penetrometry measuring equipment already mentioned. A graphical representation of the results for the PH2.3 variant of the horizontal penetrometry run is shown in Figure 6. The average value of the penetrometric resistance reached 3119.23 N (SD = 493.08 N, VK = 15.81%) in the tractor footprint and 2539.18 N (SD = 629.77 N, VK = 24.80%) off-track. The deployment of a tractor overrun with implements with a total weight of more than 10 t caused an increase in the average penetrometric resistance value on-track by more than 15%, and off-track by almost 19%. It is also clear from the graph that the measured values were higher in the wheel track than off-track (which is logical), but there were places within the measured section (150 s time period) where it was the other way around. The total number of measurements amounted to 7503 records, of which up to 21% represented higher resistance values outside the tractor wheel track. This means that up to one tractor pass is sufficient for initial compaction, and other passes may no longer have a significant effect. Statistical analysis showed that the values obtained for the on-track and off-track measurements proved to be significant ($p < 0.05$). The examination of the three 50 s sections separately also showed the dependence of tractor passage on soil compaction, with the least influence being noticeable in the middle section (F = 552.1 > Fcrit). The number of values in which the resistance value was higher outside the track than on the track of the tractor wheels was 29.59%. Statistical evaluation of the data within sections PH1.1 and PH2.3 also confirmed the hypothesis of the dependence of penetrometric resistance on the number of passes. Thus, the results obtained in the first two lines suggest that further measurement within section PH2 will therefore be of interest ($p > 0.05$). For comparison, it should be noted that

in section PH1.2, a Zetor Forterra tractor with measuring device was used for crossing twice, and in crossing PH2.3, it was also a combination of two tractors, a Valtra tractor with tools and a Zetor tractor with the measuring device. As the weight of the Valtra tractor was higher, we expected a higher compaction effect, which was confirmed, as the increase in the force required to cut through the soil on-track was 138.91 N higher. Statistical analysis showed that the tractor pass using the Valtra T202 tractor proved significant ($p < 0.05$) for the wheel track measurements. Interestingly, the result was shown to be several times more significant for the off-track investigations, whereby no wheel pass was carried out in this line. Thus, the results of the second line are also related to the input conditions, which were not the same (mean values of penetration resistance off-track for PH1.2 and PH2.3—Table 2, F = 20,632 off-track tractor, F = 294.78 on-track created by the tractor wheel).

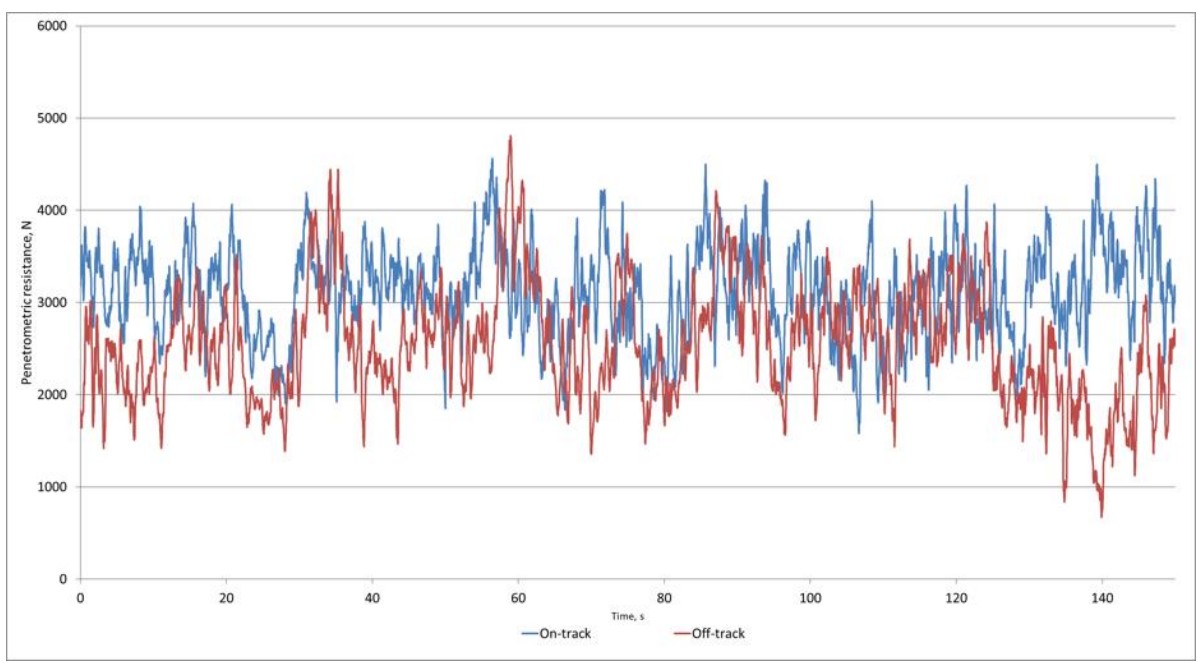

**Figure 6.** Comparative measurement, PH2.3, conditioned.

**The PH2.4 measurements** were characterized by two passes of a Valtra T202 tractor in combination with the already mentioned work tools and two passes of a Zetor Forterra tractor, where the penetrometric resistance was measured during the second measurement. The type of tire and the working pressure for the above tractor was as in measurement PH2.3. A graphical representation of the results for the PH2.4 variant of the horizontal penetrometry run is shown in Figure 7. The results show that extremes occurred in the tractor wheel tracks during the measurements (minimum value of 24.66 N—at 15 measurement points lower than 50 N, maximum value of 8919.50 N—at five measurement points higher than 8 kN). This was due to the extremely compacted clay soil strip located in this part of the parcel; the following soil analysis showed that the presence of clay at this point was more than 75%. This part of the experimental plot is very difficult to work mechanically, and the fuse element of the measuring equipment was overcome during the measurement. For this reason, it was necessary to reduce the number of measurements as the measurements with the damaged instrument were not considered relevant. The average value of the penetrometric resistance was 3180.87 N (SD = 627.32 N) with a coefficient of variation value of 19.72% for the measurements in the footprint. A significant reduction in penetrometric resistance was observed for measurements outside the tractor wheel track, the average value was 1685.17 N (SD = 465.52 N, VK = 27.62%), which represents a reduction in resistance value of up to 47%. From the statistical analysis of the effect of passes and soil compaction on the increase of penetrometric resistance, the on-track and off-track of the tractor proved to be significant ($p < 0.05$). The number of values in which

the opposite effect was shown, where the out of track values were higher than on-track was 170, which represents 2.82% of the total record. When the changes in penetrometric resistance were statistically evaluated with the addition of additional tractor passes (+1 Valtra T202 pass and one Zetor Forerra pass, for a total of four passes), the on-track line of tractor wheels proved to be significant ($p < 0.05$, F = 67.32). When evaluating the passes outside the tractor wheel tracks, it also proved to be significant ($p < 0.05$, F = 9767). From the results, it can be concluded that the movement of farm equipment over the land should be restricted to limit its compaction, thereby increasing its fertility.

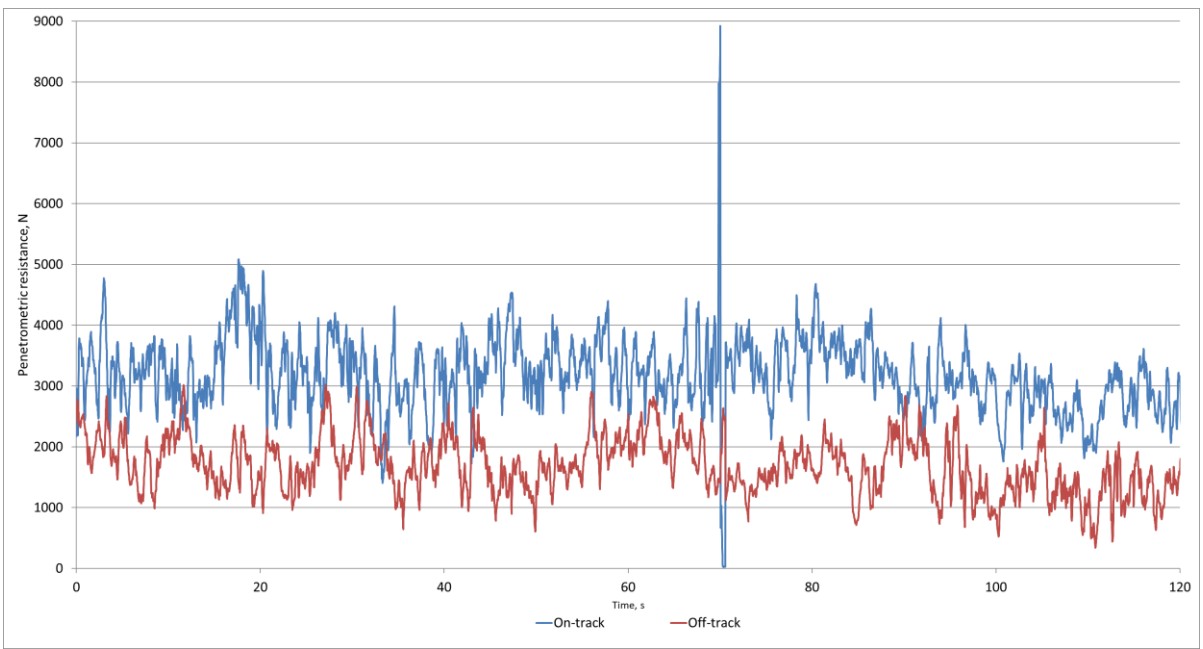

**Figure 7.** Comparative measurement, PH2.4, conditioned.

**Another pair of measurements (PH3.5 and PH3.6)** was performed under altered input conditions (Figure 1). Again, the measurements were shifted by a few meters within the surveyed terrain, and the average soil moisture content in this pass was 15.61%. The difference from the second pair of measurements was that a single pass of a Valtra T202 tractor with Kverneland Qalidisc discs was also first applied to the given research route PH3.5, changing the tire pressure to 0.15 MPa. The tires remained unchanged. Subsequently, the Zetor Forterra tractor was driven again in combination with the horizontal penetrometry measuring device already mentioned. A graphical representation of the results for the PH3.5 variant of the horizontal penetrometry run is shown in Figure 8. The average value of the penetrometric resistance was 2778.87 N in the tractor footprint (SD = 572.76 N, VK = 20.61%) and 1585.75 N off-track (SD= 521.19 N, VK = 32.87%). Deploying one tractor pass while changing tire inflation caused a 10.91% reduction in horizontal penetrometric resistance. A comparison of the results obtained for sections PH2.3 and PH3.5 (difference in Valtra tractor tire inflation) also confirmed the hypothesis that compaction would also depend on tire inflation (in the tractor wheel track: $p < 0.05$, F = 1521.22; off-track: $p < 0.05$, F = 10,206.29). A reduction in tire inflation pressure of 0.04 MPa was found to be a positive result. The graphical evaluation of the results, where 7503 data records were collected, also shows that on-track, the resistance values were higher than off-track, up to 294 values (3.92% of all measurements). The number of values above-mentioned decreased rapidly compared to the PH2.3 measurements (21% for PH2.3). A comparative examination of the achieved differences between the measured values on and off the tractor wheel footprint track revealed that values above 2 kN were found in 12.53% of the records and above 1.5 kN in 32.31% of the records. The maximum value on-track was more than 4.50 kN, while a resistance value above 4 kN was found in 1.11% off-track. The minimum value on-track

reached 1007.75 N, with a resistance less than 2 kN in 8.49% of the records. The histogram of the measured data further shows that the highest representation was in the range of 2 to 3.5 kN (90%). The obtained on-track and off-track measurements indicated the significance of the results obtained for the effect of passes on compaction ($p < 0.05$). From the measured results outside the tractor tracks in the PH2.3 and PH3.5 lines, the statistical evaluation indicated a dependence ($p < 0.05$). When compared with the measurements within line PH1.2, where two passes of the Zetor Forterra tractor were thus used in combination with the measuring equipment (1× without measurement—raised instrument, 1× with measurement), it was found that even with the application of a tractor with a higher total weight but with different tires and inflation, the average value of the penetrometric resistance was reduced by 6.76%. Thus, the statistical results indicate the significance of decreasing the tire pressure ($p < 0.05$, F = 528.55) because the mean resistance value increased when the tractor was changed (Valtra versus Zetor), but decreasing the pressure produced positive results, namely a decrease in the mean resistance value. Interestingly, values were found outside the tractor wheel tracks, where a statistical evaluation showed a dependency within the PH1 and PH3 lines ($p < 0.05$).

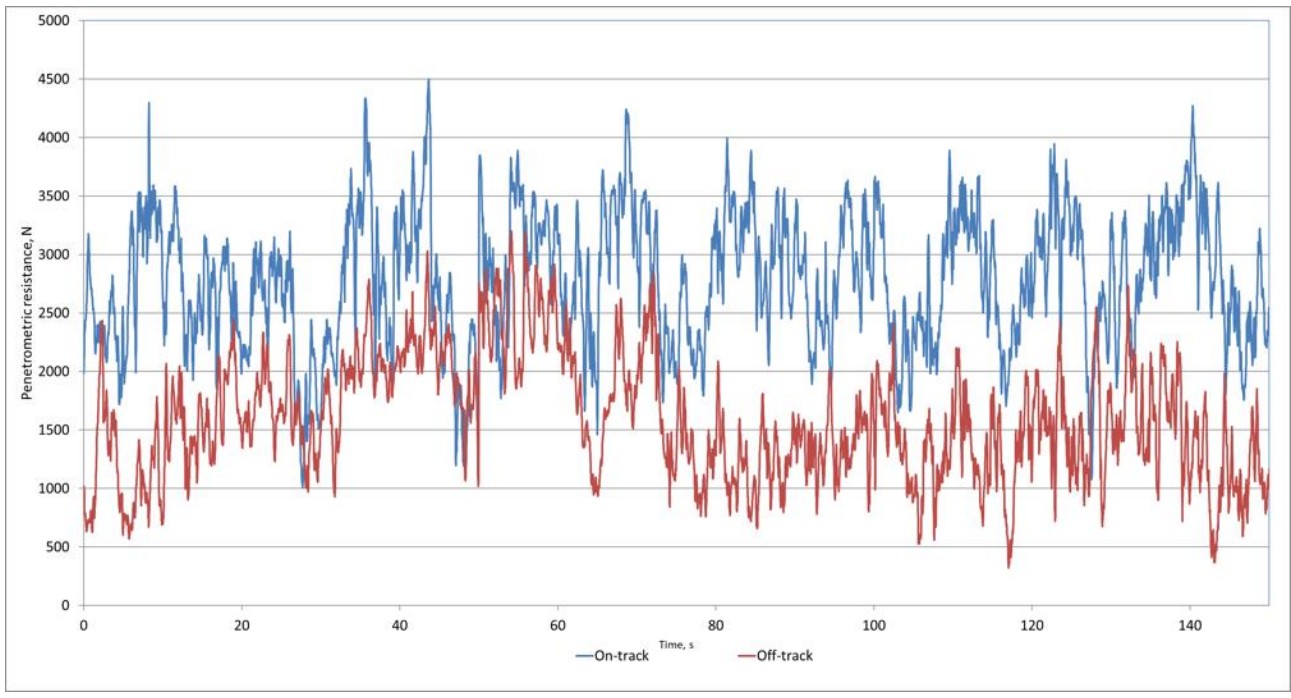

**Figure 8.** Comparative measurement, PH3.5, subconditioned.

**The PH3.6 measurements** were characterized by two passes of a Valtra T202 tractor in combination with the already mentioned work tools and two passes of a Zetor Forterra tractor, where the penetrometric resistance was measured during the second measurement. The tire type and working pressure on that tractor was the same as in the PH3.5 measurements. The change was therefore that we moved up a little from the PH2.4 measurements and changed the tire inflation pressure on the Valtra T202 tractor to 0.15 MPa. A graphical representation of the results for the PH3.6 variant of the horizontal penetrometry run is shown in Figure 9. The results show that no extremes occurred on-track during the measurements. The average value reached 2746.84 N (SD = 604.4 N, VK = 22%) on-track, while the maximum value reached 4472.84%. Off-track, the average value was 1413.31 N (SD = 574.11, VK = 40.62%), and the maximum value off-track was 3253.08 N. The results show that, when compared comparatively, up to 6.72% of the penetrometric resistance records in the area off-track were higher than those on-track. Overall, within the measured section of PH3.6, only 1.23% of the values measured on-track were greater than 4 kN, and up to 88.96% of the measured values on-track were in the range of 2 to 3.5 kN. Statistical

analysis of the effect of passes on the increase in penetrometric resistance showed that the results on-track proved to be significant with respect to the values off-track ($p < 0.05$). When comparing the comparative records obtained from the last measurements of PH3.5 and PH3.6 (i.e., after two passes of the Valtra T202 tractor and two passes of the Zetor Forterra tractor), we do not support the claim that an additional pass will cause more compaction. The average value of penetrometric resistance was even lower, by 32.02 N. The number of values that showed a negative difference (an additional pass of the tractor in PH3.6 reduced the resistance value compared to PH3.5) was more than 50% (52.74%). Thus, from the results of the univariate analysis, they do not support the hypothesis that the number of passes increases soil resistance/compaction. This phenomenon could not only have been caused by the field conditions, but also by previous measurements in PH3.5. When the measured sections off-tracks were evaluated, there was also a decrease in penetrometric resistance, namely by 172.44 N. This could also have been caused by previous measurements in section PH3.5.

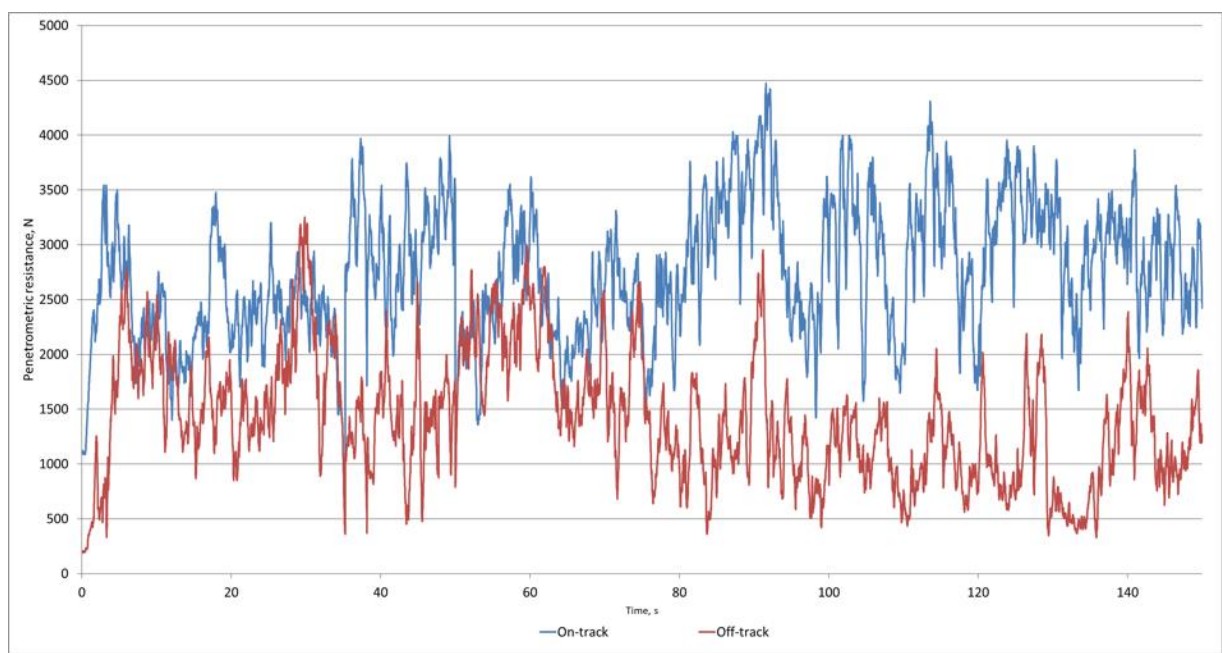

**Figure 9.** Comparative measurement, PH3.6, conditioned.

Overall, the results obtained show that soil compaction was statistically demonstrated in our investigations by the passes in our technique. Given that the three sections PH1, PH2, and PH3 were in close proximity to each other, it was considered that the soil conditions would be the same after the soil had been worked to a depth of 15 cm. The first measurement, PH1.1, was taken as a benchmark, where one pass of a Zetor Forterra tractor was made. The difference between the average value in and out of the tractor footprint was 578.03 N. The second measurement was supplemented by another pass of the same tractor, and this resulted in an increase in the average penetrometric resistance value of 9.87%. In the second measurement, the differences in the average values on-track and off-track reached 1750.18 N. The third measurement deployed a Valtra T202 tractor (1 pass) and a composite of one pass of a Zetor Forterra tractor. The difference in the on-track and off-track penetrometric resistance was 580.05 N. There was an increase in resistance of 14.99% compared to the reference value. The fourth measurement was supplemented by another pass of both tractors, and it was the line of the Valtra T202 tractor with the attachment disk, at a tire pressure of 1.9 MPa, that caused the greatest compression as well as the greatest increase (1495.70 N—the difference between on-track and off-track resistance). Compared to the reference value, there was an increase in the average horizontal penetrometric resistance of up to 17.26%. The fifth measurement was

set to the same conditions as the third, but with the difference that the tire pressure was reduced by 0.04 MPa for the Valtra T202 tractor. The difference between the on-track and off-track resistance was 1193.12 N, while the increase in resistance relative to the reference value was low at 2.44%. The last measurement (sixth) was set as the fourth, but again with underinflated tires (0.04 MPa reduction). This measurement recorded a difference on-track and off-track of 1333.53 N. Overall, the increase in the average resistance value relative to the reference value was minimal (1.26%). From these results it can be said that by simply reducing the tire pressure, it is possible to reduce the soil compaction by up to 16% (results of the sixth and fourth measurements).

When comparing the individual passes (PH1.2, PH2.4, and PH3.6), the statistical analysis showed that the results for the horizontal resistance measurements were statistically significant ($p < 0.05$) with respect to the weight, number of passes, and underinflation of the tires. For the tractor wheel track measurements, the results showed that the first pass of the Zetor tractor (PH1.1) caused the greatest compaction, while the next pass (PH2.3) with the heavier Valtra tractor only increased the compaction by 14.19%. The pass in section PH3.5 with under-inflated tires (Zetor tractor and Valtra tractor passes) caused the compaction to increase by only 1.73% compared to the PH1.1 pass.

When comparing the individual passes (PH1.2, PH2.4, and PH3.6), the statistical analysis showed that the results for the horizontal resistance measurements proved statistically significant ($p < 0.05$) with respect to the weight, number of passes, and tire underinflation. For the tractor wheel track measurements, the results showed that the first two passes of the Zetor tractor (PH1.2) caused the greatest compaction, while the next pass (PH2.4) of the Valtra tractor (higher tractor weight) only increased the compaction by 8.99%. The PH3.6 passes in the PH3.6 line with undercompacted tires proved to be significant in reducing the soil compaction, as on average, the drag values were more than 10% lower (compared to PH2.4).

## 4. Discussion

Soil compaction is one of the main negative factors that limit plant growth, crop yield, and increases the runoff of water from the land due to the deterioration in the soil's ability to absorb water. Therefore, it is important to determine the level of soil resilience and map it for the field to find solutions to the negative effects of compaction [12]. Soil compaction is an important physical limiting factor for root growth and plant emergence and is one of the major causes for reduced crop yield worldwide, thus determining the type and degree of tillage that suit the soil conditions [13]. Soil compaction may significantly debilitate the production capacity of soil by reducing porosity, creating obstacles to air, water, nutrient movements, and root penetration [14]. The reduction in the yield of selected crops can be attributed to soil compaction even in the study of different soil types, from sand to clay [15,16]. Soil compaction measurements have been addressed by several authors. Penetrometric resistance and overall compaction will depend on the soil moisture at the moment and the number of passes of the equipment [17]. In addition to soil moisture, other essential properties include texture, method of cultivation, bulk density, and organic carbon content [18]. The passage of technology causes a deterioration in the physical properties of the soil, especially an increase in bulk density, which causes excessive soil compaction [19]. The measurement of penetrometric resistance is usually carried out by vertical measurement. This, as we know, poses a certain problem for fast and large scales [20]. In other measurements, the failure mode of the soil has been investigated by two horizontally operated penetrometers. The results show that at small depths, the differences were more evident between the horizontal and vertical measurements. At greater depths, similarities in compaction were demonstrated [21]. The operation of agricultural machinery can cause soil compaction and a high variability in soil structure. Penetrometer resistivity changes were irregular at the surface but showed high spatial correlation between data measured at different depths. A penetrometer-measured resistance exceeding 2.5 MPa is critical for root growth. In certain measurements, it increased from 20% to 40% at a

probability level of $p > 0.40$ after five passes of the tractor [22]. From a practical point of view, it appears to be the best and simplest comparative method to obtain objective results. This method involves the continuous measurement of two parameters. Based on this principle, a measuring device with a two-argument comparison method has been developed at the Institute of Agricultural Engineering, Transport. and Bioenergy [11]. Reducing compaction can also be achieved by changing tires [23].

The design of a penetrometer that can make instantaneous measurements of horizontal soil resistivity in the soil and global positioning system (GPS)-based data collection software has also been developed by other authors that can map the soil resistivity down to a depth of 40 cm. The measurements resulted in soil resistance values ranging from 0.2 MPa to 3 MPa. In conclusion, the experimental results showed that the proposed system works quite well in the field, and a horizontal penetrometer is a practical tool to provide online soil resistivity measurements [12].

Many studies in the scientific literature point to the importance of immediately determining the level of soil resilience and mapping its properties. To achieve this, it is necessary to focus on the design of penetrometers that can make instantaneous measurements [24,25].

Automatic oscillations were caused by the tractor and measuring equipment, the nonuniformity of soil properties, potential nonuniformity of the depth of previous soil cultivation, and nonuniformity of the soil surface. The distance from the row to the previously cultivated crop were also not taken into account in the measurement as it was not visible due to stubble cultivation. These factors, together with the soil variability, reflect realistic soil conditions at work in crop production, but may also have influenced the accuracy of the measurements and thus the results. For all our measurements, the same setting was used: the measuring instrument as well as the three-point linkage setting and a constant movement speed. For possible further investigations with this type of measuring device, we recommend taking the above-mentioned factors into consideration.

## 5. Conclusions

An important input factor was that there was a long-term rainfall deficit in the period before and during the experimental measurements, which corresponded to the measured values for soil moisture. The overall average soil moisture was 17.36%. In the framework of the investigations during the deployment of agricultural work according to the controlled movement of machinery in the field, we were able to carry out measurements of horizontal penetrometry with the following hypothesis evaluations obtained:

➢ The passage of machinery causes soil compaction;
➢ Multiple passes of equipment increases overall compaction, with some exceptions;
➢ Reducing the tire pressure of the tractor tires will reduce the impact of tractor passes on compaction.

Horizontal penetrometry with a measuring device designed at the Institute of Agricultural Engineering, Transport, and Bioenergetics was evaluated for six different variants of tractor passes with implements or measuring equipment. The measurements and results indicated a possible increase in resistance from average values of 1230.14 N to 3180.87 N. However, the maximum values were far higher in places and exceeded 4.5 kN.

An interesting conclusion of the measurement was also the finding that reducing the tractor tire pressure to 0.015 MPa resulted in a reduction in soil compaction of up to 16%.

During one measurement, there was also a failure of the measuring equipment where there must have been an obstruction in the soil that caused the safety element to shear. The equipment proved to be suitable for the rapid measurement and mapping of the compaction status of the land, with the results serving the farm as a guide for the future planning of work operations with a selection of equipment aimed at reducing the development of further compaction.

**Author Contributions:** Conceptualization, M.M. and J.J.; Methodology, M.M. and J.K.; Software, M.M.; Validation, M.M. and J.J.; Formal analysis, M.M., J.K. and J.J.; Investigation, M.M. and J.J.;

Resources, M.M.; Data curation, J.J.; Writing—original draft preparation, M.M., J.K. and J.J.; Writing—review and editing, M.M. and J.J.; Visualization, J.K. and J.J.; Supervision, M.M.; Project administration, M.M. and J.J.; Funding acquisition, M.M., J.K. and J.J. All authors have read and agreed to the published version of the manuscript.

**Funding:** This research received no external funding.

**Data Availability Statement:** No new data were created or analyzed in this study. Data sharing is not applicable to this article.

**Acknowledgments:** We greatly thank the editor and reviewers for their valuable comments to improve the quality of this paper.

**Conflicts of Interest:** The authors declare no conflicts of interest.

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
