# Peer review of "Comparative Measurement of Horizontal Penetrometry with a Focus on the Degree of Soil Compaction in Real Work Conditions"

_agriengineering, doi:10.3390/agriengineering6020056_

Round 1
Reviewer 1 Report
Comments and Suggestions for Authors
In the context of continuous soil compaction caused by mechanized tillage, this paper uses a new device to measure soil horizontal resistivity to measure soil compaction, by changing the weight of the tractor, the number of trips, and the tire pressure, measuring and comparing the soil horizontal resistance on and off the tire track in three 50s sequences, and performing a one-way ANOVA on the results, it is believed that appropriately reducing the tire pressure and the number of walks is conducive to reducing the soil compaction. Different from the traditional vertical measurement method, the author uses the horizontal measurement method to measure soil compaction, which is innovative, and the conclusion has certain guiding significance for the scientific cultivation of farmland in the future. However, there are some shortcomings and corresponding suggestions in the paper:
1. The title should also emphasize that the measurement results measure the degree of soil compaction.
2. The conclusions written in the abstract should be appropriately added.
3. The serial numbers of the figures in the paper are incorrect, all of them are figure 2.
4. Consider adding a transverse single-factor comparison test on 3 paths, and the factors are weight, number of walks and tire pressure.
5. Lack of prospective content, such as increasing the comparative test of black soil and clay loam, and the enlightenment of future farming path planning.
6. More references need added and cited.
Comments on the Quality of English LanguageMinor editing of English language required
Reviewer 2 Report
Comments and Suggestions for Authors
This manuscript is of scientific and practical interest. This study presents a proposal to investigate the effect of the number of passes of wheeled machinery, its weight and tire pressure on soil retention.
The paper is presented in a clear manner with sufficient data to understand the proposed solution. There is a sufficient overview and information on the relevance of the problem under study. However, there is no information about the size of the real contact patch of tractor tires depending on the type of tires and tire pressure.
In the process of experiment and at the analysis of results the dependences characterizing specific resistance of soil during the time of movement are given, however on the given resistance will influence the instantaneous speed of movement of a stylus of horizontal penetrometer, on which will influence irregularity of movement of a tractor, rigidity of construction of system of fastening of a stylus, auto oscillations created by system of measurement, irregularity of properties of soil on length of measured area, irregularity of depth of cultivation of soil, irregularity of a surface of soil. At present, there is no description and consideration of these factors in the research. Also the distance from the row of the previously grown crop will have a significant influence on the measurement process, as the root system will influence the structure and properties of the soil and if one stylus goes along the row (or near the row) and the other in the middle of the inter-row - their resistance values will differ significantly.
Figure 2 (a, b) shows the measuring device, but there is no description or design. How was it calibrated? What types of sensors were used and with what degree of accuracy were measurements made?
Depending on the loading, the soil compacts to form a rut. How were measurements made - from the original surface to a depth of 20 cm or from the surface of the layer compacted by the tire?
In Figures 4-9 it is recommended to remove two zeros after the decimal point on the vertical axis.
It is also not clear from the study how reducing tire pressure will affect fuel consumption, tire durability and subsequent economic effect.
At the same time, the methodology of the conducted research and the description of the results are well enough stated in the article, which is a significant plus of this publication.
The conclusions obtained as a result of the work reflect the results of the conducted research.
This work is theoretical and practical in the future can be used by other scientists and subsequently improved, as well as taken as a basis for the study of traction characteristics of new tillage machines and to justify the parameters of pressure in the tires of different tractors when performing various operations.
Reviewer 3 Report
Comments and Suggestions for Authors
The issue of horizontal penetrometer measurement is a current research need.
The authors chose the measurement procedure with the potential to achieve relevant scientific results.
The authors studied a sufficient number of literary sources related to the chosen issue but it is recommended to add other sources related to the chosen issue to the references.
The level of the manuscript can be increased by recommendations regarding the addition of soil moisture data:
-When measuring with a penetrometer, the current soil moisture is of fundamental importance. The long-term rainfall deficit in the period before and during the experimental measurements is serious.
-Soil moisture is probably given as a percentage by weight. It is recommended to indicate soil moisture also in percent of volume.
Answers to the questions the authors asked themselves when planning the field experiment are well summarized in the conclusions.
Tables and figures - it is advisable to add the required data on soil moisture.
